

# Core body temperature correlates of transition from aerobic to anaerobic metabolism in running

Marija Rakovac[1,*], Davor Šentija[1,*], Tošo Maršić[2] and Vesna Babić[1]

[1] University of Zagreb Faculty of Kinesiology, Zagreb, Croatia
[2] Faculty of Science and Education University of Mostar, Department for Kinesiology, Mostar, Bosnia and Herzegovina
[*] These authors contributed equally to this work.

## ABSTRACT

**Purpose.** We investigated core body temperature (CBT) during a graded exercise test (GXT) in comparison with gas exchange dynamics.

**Methods.** Thirty-two active males performed a treadmill GXT (0.5 km/h increments every 30 seconds, 1.5% incline) until exhaustion. Gas exchange data and rectal temperature ($T_{re}$) were continuously registered. Ten participants repeated the test for reliability assessment. The first and second gas exchange thresholds ($VT_1$ and $VT_2$) were determined by the simplified V-slope method, while CBT dynamics and eventual temperature thresholds ($TT_1$ and $TT_2$) were assessed according to the criteria defined in this study. Three independent evaluators determined gas exchange and temperature thresholds.

**Results.** In 29 subjects, $T_{re}$ increase was best fitted with a 3-phase segmented model of successively steeper slopes, with a linear relationship in all three segments (17 subjects), or in two segments, with a quadratic relationship for the remaining segment (12 subjects). The between-segment intersection points were considered as $TT_1$ and $TT_2$. In three participants, $T_{re}$ was best fitted with a two-segment, single-breakpoint ($TT_1$ or $TT_2$) model. The evaluators' objectivity was satisfactory for $VT_1$ ($\alpha = 0.786$), very high for $TT_2$ ($\alpha = 0.941$) and $VT_2$ ($\alpha = 0.948$). $TT_1$ and $VT_1$ were moderately correlated ($\rho = 0.41$, $p = 0.021$) while $VT_2$ and $TT_2$ were highly correlated ($r = 0.78$, $p < 0.001$) showing a small, yet statistically significant difference ($12.95 \pm 1.9$ vs $13.43 \pm 1.7$ km/h, $p = 0.039$). However, test-retest reliability was low.

**Conclusion.** The breakpoints in CBT increase observed during graded running may represent transitions between the three intensity domains of physical activity.

## INTRODUCTION

All physical activity can be sorted into three distinct intensity domains—moderate, heavy, and severe (*Skinner & Mclellan, 1980*; *Antonutto & Di Prampero, 1995*; *Meyer et al., 2005*; *Binder et al., 2008*). They reflect different metabolic sources and processes providing energy for muscular contraction, and are delineated by two transition points, or thresholds. The

Corresponding author
Marija Rakovac,
marija.rakovac@kif.unizg.hr

aerobic (or lactate) threshold (AeT) is the upper limit for the moderate intensity. Exceeding the AeT intensity, the body is progressively more reliant on carbohydrates *vs* fats as fuel, and blood lactate [La -] increases significantly compared to resting values. Carbon dioxide output ($VCO_2$) and pulmonary minute ventilation (VE) show disproportionate increase compared to oxygen uptake ($VO_2$), needed to eliminate excess $CO_2$ produced by working muscles, and maintain homeostasis of $CO_2$ pressure (*Wasserman et al., 1973*; *Kindermann, Simon & Keul, 1979*; *Meyer et al., 2005*). The upper limit for the heavy domain is the anaerobic threshold (AnT), also called respiratory compensation point (RCP) (*Wasserman et al., 1973*). AnT represents the highest intensity at which there is still an equilibrium between lactate production and elimination (maximal lactate steady state), and at higher intensities blood [La −] and [H +] rise inexorably and cannot be stabilized. At the AnT there is a second disproportionate increase in $VCO_2$ and VE as compared to $VO_2$, as well as a decrease in arterial pH and $CO_2$ pressure (*Rusko et al., 1986*; *Meyer et al., 2005*). To test whether buffering of the metabolic acidosis prevents the occurrence of the AnT, *Meyer et al. (2004)* administered intravenous injections of bicarbonate during a ramp cycling test. The delayed (but not prevented) occurrence of AnT in their study indicates that exercise induced metabolic acidosis is, to a certain extent, causally involved in the occurrence of hyperventilation at the AnT.

This concept of three domains and two threshold intensities as delineating breakpoints has long been recognized and used both in medicine and sports (*Loat & Rhodes, 1993*)—in prescription of exercise intensity, monitoring of training and rehabilitation effects, performance and clinical outcome prediction, and athlete selection (*Meyer et al., 2005*; *Faude, Kindermann & Meyer, 2009*). Despite its widespread use, the concept of aerobic/anaerobic thresholds still remains a controversial topic in terms of its definition, underlying physiological mechanisms, as well as its measurement methods and even its mere existence (*Brooks, 1985*; *Anderson & Rhodes, 1989*; *Loat & Rhodes, 1993*; *Myers & Ashley, 1997*; *Bosquet, Léger & Legros, 2002*; *Svedahl & MacIntosh, 2003*; *Poole et al., 2021*).

In addition to the most widely used methods of threshold identification—blood lactate and gas exchange measurements (*Svedahl & MacIntosh, 2003*; *Meyer et al., 2005*)—a number of other methods have been proposed to explain and enhance the objectivity of the determination of aerobic-anaerobic transition. These include field and laboratory measurements of heart rate (*Conconi et al., 1982*), heart rate variability (*Kaufmann et al., 2023*), catecholamines in plasma (*Schneider, McGuiggin & Kamimori, 1992*; *Chwalbinska-Moneta et al., 1996*), myoelectric signals (*Nagata et al., 1981*; *Lucía et al., 1999*), ammonium ion in plasma (*Buono, Clancy & Cook, 1984*; *Yuan et al., 2002*), saliva electrolytes (*Chicharro et al., 1994*) or self-reported subjective measures (*Bok, Rakovac & Foster, 2022*).

Different research approaches reflect different hypotheses on the physiological mechanisms underlying the thresholds. The use of electromyography (EMG) to analyze the aerobic-anaerobic transition during graded exercise (*Nagata et al., 1981*; *Moritani et al., 1984*; *Airaksinen et al., 1992*; *Mateika & Duffin, 1994*; *Chwalbińska-Moneta, Hanninen & Penttila, 1994*; *Taylor & Bronks, 1994*; *Lucía et al., 1997*; *Lucía et al., 1999*; *Hug et al., 2003*; *Frazão et al., 2021*) is based on the hypothesis that the increases in EMG activity, resulting from the increased recruitment of fast twitch motor units during incremental exercise,
would appear as noticeable thresholds that would correlate with the ones detected by conventional methods for determination of the aerobic-anaerobic transition.

Building on this, we formulated a hypothesis on possible temperature thresholds based on the changes in motor units recruitment during incremental exercise. Namely, it has been shown that within muscle fibers, compared to type-I isoforms, type-II specific myosin ATPase isoforms require $1.6-2.1$ times more ATP per unit force production and therefore have a proportionately lower thermodynamic efficiency (*Stienen et al., 1996*; *Reggiani, Bottinelli & Stienen, 2000*; *Barnes & Kilding, 2015*). Moreover, an increase in intracellular free ADP, $P_i$ and $H^+$ occurs with increasing intensity of muscle contraction, leading to a decrease of the free energy ($\Delta G$) for ATP hydrolysis (*González-Alonso et al., 2000*). Therefore, we hypothesized that a more pronounced recruitment of fast twitch motor units, and lower thermodynamic efficiency characteristic for increasing intensity of exercise and the aerobic-anaerobic transitions, would reflect in clearly detectable threshold-like changes in the increase of the core body temperature (CBT) during a ramp test in controlled environmental conditions. As the recruitment of fast-twitch motor units (especially type IIb at the intensities above the anaerobic threshold) results in an appreciable increase in the energy turnover per unit of work, a disproportionate increase in heat production might be expected, and detected as threshold-like CBT changes. Consequently, we hypothesized that with increasing exercise intensity the CBT slope changes would coincide with the changes in ventilatory parameters used to determine the first (aerobic) and the second (anaerobic) gas exchange thresholds ($VT_1$ and $VT_2$, respectively).

In the context of ventilatory and temperature parameters, *Whipp & Wasserman (1970)* were the first to explore the relationship between hyperpnea and rectal temperature during a progressive exercise test to fatigue, in normothermic and hypothermic state. Pulmonary ventilation in their study increased proportionally to $VCO_2$ and was independent of the body temperature level, suggesting that during exercise body temperature cannot be considered an independent stimulus to ventilation. *White & Cabanac (1995)*; *White & Cabanac (1996)* also explored the ventilatory response as a function of body temperature increase during incremental cycling. Contrary to the study of *Whipp & Wasserman (1970)*, they (*White & Cabanac, 1995*; *White & Cabanac, 1996*) reported the occurrence of a breakpoint in the relationship between esophageal and tympanic temperatures and ventilatory equivalents for $VO_2$ and $VCO_2$, thus termed as core temperature threshold for increased ventilation (*White & Cabanac, 1995*; *White & Cabanac, 1996*). They assumed that hyperventilation during high-intensity exercise could be in part a thermolytic response involved in selective brain cooling.

Countless graded exercise tests are performed each day in clinical and sports diagnostics settings. Remarkably, to the best of our knowledge, despite the importance and extensive research of the CBT and thermoregulation, CBT dynamics in relation to increasing work rate have not been addressed in prior studies. Moreover, none of these studies evaluated core body temperature and its relationship with gas exchange data during a finely graded running test to exhaustion. Thus, the aim of our study was to (1) explore and characterize core body temperature response to graded treadmill exercise, and (2) compare core body temperature and gas exchange data measured concurrently during

a finely graded GXT. Rectal temperature ($T_{re}$) has been shown as a reliable and stable measure/representative of the internal body temperature (*Armstrong et al., 2007*; *Lee et al., 2010*) and it was intentionally chosen as the index of CBT in our study due to its proximity to the large muscle groups active in running. We hypothesized that with increasing exercise intensity the CBT would show a disproportionate increase (with threshold-like phenomena) coincident with the changes in ventilatory parameters used to determine the aerobic and the anaerobic thresholds.

## MATERIALS & METHODS

### Participants

Thirty-two recreationally active males (age 26.5 (6.6) yrs; height 179.2 (5.0) cm; body mass 76.8 (8.1) kg; mean (SD) for all values) participated in the study. The participants were kinesiology students and recreational runners recruited by the Diagnostic Center of the University of Zagreb Faculty of Kinesiology. The inclusion criteria were engagement in running activities at least four hours per week during the year prior to testing, experience in treadmill running, absence of any musculoskeletal or cardiovascular symptoms or diseases that might have influenced the results, normal resting body temperature on the day of the testing, and no recent exposure to high environmental temperatures. The measurements were performed during the spring and autumn months, so no seasonal acclimatization was expected. The participants were asked to refrain from strenuous physical activity at least 24 h, have a light breakfast at least 2 h before testing, and show up dressed in light clothing and running shoes. The purpose of the study and potential harms were explained, and written informed consent provided from all subjects. The study, approved by the Review Boards of the University of Zagreb Faculty of Kinesiology and School of Medicine (approved research proposal 04-3741/2-2008), conforms to the Helsinki Declaration.

### Study protocol

All subjects performed a graded exercise test on a motorized, calibrated treadmill (Run Race, Technogym, Italy) with incline set at 1.5%. Following a short warm-up and stretching procedure, the participants started walking for 3 min at three km/h. Thereafter, speed increased by 0.5 km/h every 30 s. The participants were instructed to start running at the speed of five km/h. At low running speeds the aerial (airborne) phase was not required, as just the stance leg had to be flexed in the mid-stance phase (spring-mass model). The test was performed until volitional exhaustion. The last full stage the subject could sustain was defined as the subject's maximal speed. During recovery, the subjects walked at five km/h for 5 min. To determine the test-retest reliability, ten participants repeated the test within one week.

All tests were performed in the morning hours, in stable microclimatic conditions (air temperature 20−22 °C, relative humidity ≤ 60%, no appreciable sources of air flow) to keep the increase of the CBT proportional to the amount of metabolically produced heat (*Lind, 1963*).

## Acquisition and analysis of gas exchange parameters

We monitored gas exchange parameters continuously throughout the test, using an automated breath-by-breath system (Quark $b^2$, Cosmed, Rome, Italy). The system includes a face mask worn by participants (Hans Rudolph, Shawnee, KS, USA), connected to a turbine with opto-electronic air flow meter. Expired air (one ml/s) passes through a capillary tube (*Nafion Permapure®*) to reach analyzers for $O_2$ (zirconia) and $CO_2$ (infrared). The gas concentrations are measured with an accuracy of $\pm\ 0.03\%$. Prior to each test the turbine was calibrated with a 3-L pump, and the $O_2$ and $CO_2$ analyzers were calibrated with a calibration gas ($O_2$ 16.10%, $CO_2$ 5.20%, $NO_2$ rest).

Gas exchange data analysis and $VT_1$ and $VT_2$ assessment were performed within the Quark $b^2$ software (Version 8.1a, Cosmed, Italy) as previously described (*Sentija & Markovic, 2009*). Briefly, determination of both thresholds is based on the accelerated rate of $CO_2$ output compared to $VO_2$ (simplified V-slope method) (*Sue et al., 1988*; *Schneider, Phillips & Stoffolano, 1993*). The point of the first disproportionate increase of $VCO_2$ in comparison to $VO_2$ indicated the oxygen uptake at $VT_1$ and the corresponding speed. The point of the second disproportionate increase of $VCO_2$ in comparison to $VO_2$ (the intersection point of the below and above regression lines) represented the second ventilatory (anaerobic) threshold ($VT_2$). When needed, threshold determination was supported with inspection of the VE, VE-$VCO_2$ plot, respiratory exchange ratio (RER), and ventilatory equivalents for $O_2$ ($VE/VO_2$) and $CO_2$ ($VE/VCO_2$). The highest oxygen uptake for any 30-s period recorded in the incremental running test was defined as peak $VO_2$. Three experienced evaluators detected $VT_1$ and $VT_2$ independently.

## Acquisition and analysis of core body temperature

Rectal temperature was measured with a single-use fast-response temperature probe connected to a datalogger (*Xplorer GLX*, PASCO Scientific, Roseville, CA, USA) with *DataStudio* software for data storage and analysis. The logger contains built-in temperature sensors with a resolution of 0.01 °C. The frequency of data collection was set at two Hz. Prior to each testing, the instrument was calibrated according to the manufacturer's instructions. Subjects were carefully instructed to insert the temperature probe into the rectum, eight cm from the anal opening (*Åstrand et al., 2003*). The probe cable was fastened to the participants' shorts and additionally secured with an elastic strap around the waist. After the test, temperature data were exported to Microsoft Excel and analyzed graphically.

If manifested, the temperature thresholds ($TT_1$ and $TT_2$) were assessed by the same evaluators that detected the gas exchange thresholds. The evaluators were instructed to determine the temperature thresholds visually (by detecting breakpoints of abrupt change in the slope of the CBT-time/speed relationship) and perform computer-assisted regression analysis for confirmation. To improve the detection of the thresholds, all data were viewed and analyzed graphically as raw data, and with three different time-averaging intervals: 15-s (half-stage), 30-s (full stage), and 60-s (two-stage). Temperature data for a subject with three distinct linear phases and clear $TT_1$ and $TT_2$ are shown in Fig. 1.

Piecewise linear and second-order polynomial (quadratic) models were used to fit temperature data. If both the linear and polynomial functions fitted the data equally well,
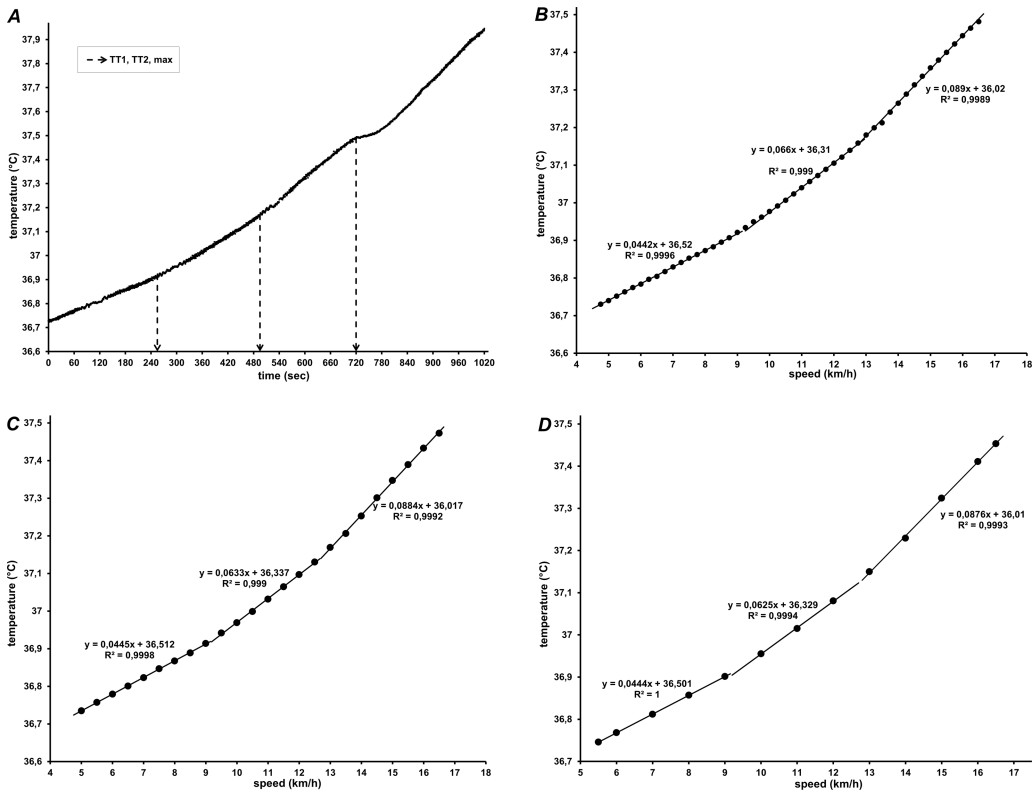

**Figure 1** Core body temperature of one participant during graded treadmill running, in relation to (A) time (raw data), and speed averaged every 15 s (B), 30 s (C) and 60 s (D). In (A) $T_{re}$ throughout 5 min of recovery is also shown. In (B, C, and D) regression lines with corresponding equations for the three segments are shown.

the lower order (linear) model was chosen. To detect trend changes and avoid overfitting, a minimum range of CBT observations for any phase was defined as four stages (2 min, or two km/h range) and a maximum number of two allowed breakpoints was selected, in line with the presumed three-domain model of physical activity. This prevented that detected breakpoints were solely affected by step-to-step changes and supported the detection of only major breakpoints in the entire intensity range. Furthermore, to identify the breakpoint as either $TT_1$ or $TT_2$ in cases where only one breakpoint could be discerned, it was assumed that both breakpoints are roughly equidistant from start to end of running in the test.

Evaluation of the temperature and ventilatory data was performed separately, and the evaluators were blinded for both data sets. If evaluators disagreed about a detected gas exchange or temperature threshold value, the value considered in further analyses was either: (a) the value on which two evaluators agreed (in case only one evaluator selected a different threshold value), or (b) the median value (in case all three evaluators selected different threshold values).

## Data analysis

Descriptive statistics (arithmetic mean, standard deviation and range) were calculated for each variable. The normality of data distribution was tested by the Kolmogorov–Smirnov test. The relationship between ventilatory and temperature parameters was tested using Pearson's correlation coefficient ($TT_2$ and $VT_2$) and the Spearman rank correlation coefficient ($TT_1$ and $VT_1$). The statistical significance of the differences between ventilatory and temperature parameters was tested by the paired samples $t$-test ($TT_2$ and $VT_2$) and the Wilcoxon signed-rank test ($TT_1$ and $VT_1$). The agreement between the temperature and ventilatory thresholds was tested using regression analysis with 95% prediction and confidence intervals and by the Bland-Altman method (*Altman & Bland, 1983*). The objectivity of the three evaluators in identification of the ventilatory and temperature thresholds was determined by the Cronbach α coefficient. The test-retest reliability was determined by the intraclass correlation coefficient (ICC). In order to reveal the dynamics of CBT in relation to running speed and in relation to gas exchange thresholds within the whole dataset the following procedure was used: (1) the relative speed for each stage was calculated (in % of maximal speed), for each participant; (2) CBT data for all participants were added together, grouped and averaged within categories representing a relative 2% running speed increase (from 24 to 100%). Statistical significance was set at $p \leq 0.05$. All statistical analyses were performed using Statistica, version 14.0.1.25 (StatSoft Inc., Tulsa, OK, USA).

# RESULTS

## Modeling of core body temperature

Different patterns characterized the increase of rectal temperature during graded treadmill running. The most common (detected in 17 participants) was a 3-phase segmented linear regression model, with successively steeper temperature slopes at higher speeds (Fig. 1). The second model was detected in 12 participants: the temperature data were best fitted with a quadratic relationship in one segment, and a linear relationship for the remaining two segments (Fig. 2). The breakpoints between the segments were considered as the first and second temperature thresholds ($TT_1$ and $TT_2$).

In the remaining three subjects, the temperature data were best fitted with a single breakpoint ($TT_1$ or $TT_2$) between two segments, one with a quadratic and one with a linear function (Fig. 3).

The mean resting rectal temperature at the start of the exercise test was $37.20 \pm 0.27\,°C$. Initial dips in rectal temperature were observed in certain subjects, but no systemic tendency at the onset of running exercise was found. That is, $T_{re}$ within the first (moderate) domain slowly rose in some, whilst it was preserved or even decreased in others. Rectal temperature rose with higher exertion, and continued to increase throughout the recovery period (see Fig. 1A), during which the highest values were observed in all subjects (on average, $38.42 \pm 0.38\,°C$).

Four participants showed a sporadic drop in temperature readings lasting 1–5 s, probably caused by errors in signal transmission; the missing values of those brief periods between

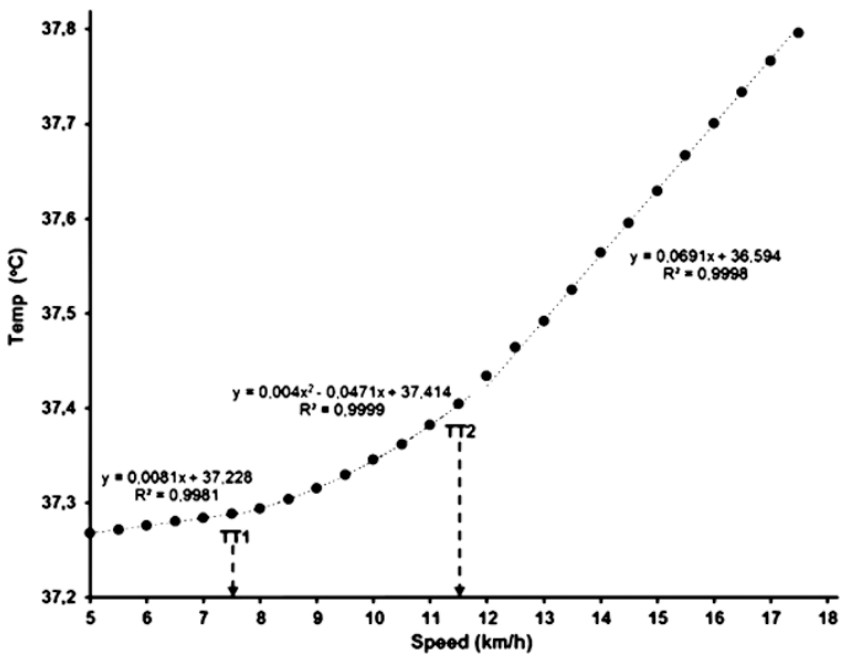

**Figure 2** Core body temperature in relation to running speed in one participant with one quadratic (central) and two linear (initial and final) segments.

validly collected temperature data were easily recovered with appropriate estimated values, allowing us to construct representative core temperature profiles throughout the measurement for all participants.

## Gas exchange and temperature thresholds

Descriptive data of the temperature and gas exchange thresholds are presented in Table 1. Both gas exchange thresholds ($VT_1$ and $VT_2$) were determined from the $VO_2/VCO_2$ relationship (V-slope, Fig. 4A). However, due to artefacts in the gas exchange data, one participant was excluded from the analysis. In three participants, there was a clear breakpoint with an abrupt rise between the first and second segment in the $VCO_2/VO_2$ relationship (and was adjudicated as $VT_1$), although the slope coefficient for the second segment (data between $VT_1$ and $VT_2$) was below 1.00 (Fig. 4B). Overall, smoother between-phase transitions and higher signal-to-noise ratio were noted for CBT, than for gas exchange parameters.

## Objectivity of the evaluators

Although noticeable in most subjects, the evaluators' objectivity in the assessment of $TT_1$ was unsatisfactory ($\alpha = 0.693$). In seven participants one of the evaluators could not identify $TT_1$. On the other hand, the evaluators' objectivity in the assessment of $TT_2$ was very high ($\alpha = 0.941$). The objectivity of the assessment of gas exchange thresholds was satisfactory both for $VT_1$ ($\alpha = 0.786$) and for $VT_2$ ($\alpha = 0.948$).

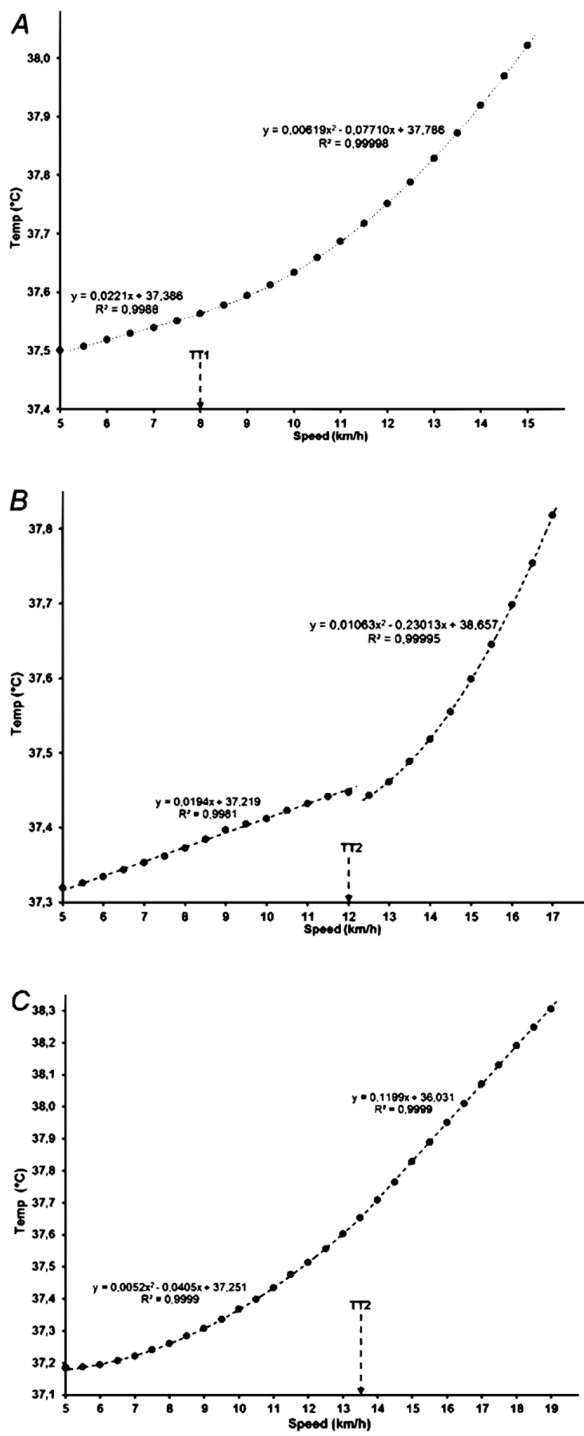

**Figure 3** Core body temperature with a single breakpoint in relation to running speed (A, B and C showing data for three participants).

**Table 1** Descriptive statistics of the variables at the thresholds, and at peak running speed.

| | Speed (km/h) | Temp (° C) | $VO_2$ (ml/kg) | $\%VO_{2max}$ | K-S p |
|---|---|---|---|---|---|
| $TT_1$ | 8.14 ± 0.9 (7–11) | 37.33 ± 0.29 (36.61–37.83) | 33.9 ± 5.0 | 56.8 ± 7.1 | <0.01 |
| $VT_1$ | 8.35 ± 1.3 (6–11) | 37.33 ± 0.29 (36.65–37.83) | 34.5 ± 4.4 | 57.6 ± 5.2 | >0.20 |
| $TT_2$ | 13.43 ± 1.7 (10.5–18) | 37.59 ± 0.31 (36.92–38.13) | 51.2 ± 6.1 | 85.4 ± 6.0 | >0.20 |
| $VT_2$ | 12.95 ± 1.9 (9.5–17.5) | 37.57 ± 0.29 (36.98–38.09) | 50.3 ± 5.3 | 83.6 ± 3.9 | >0.20 |
| Max | 17.78 ± 2.1 (14–22.5) | 37.96 ± 0.37 (37.11–38.84) | 59.9 ± 5.4 | — | >0.20 |

**Notes.**

$TT_1$ and $TT_2$, first and second rectal temperature threshold; $VT_1$ and $VT_2$, first and second gas exchange threshold; Max, maximal values at exhaustion; Temp, rectal temperature; K-S p, $p$-value in Kolmogorov–Smirnov test.

Values are means ± standard deviation, with ranges in parentheses.

## Relationship between temperature and gas exchange thresholds

No significant differences were found between $TT_1$ and $VT_1$ ($p = 0.407$), while there was a small, but significant mean difference between running speed at $TT_2$ and $VT_2$ ($p = 0.039$). However, when expressed as temperature, oxygen uptake or percentage of $VO_{2max}$, the average values of $VT_2$ and $TT_2$ thresholds did not differ significantly ($p > 0.10$ in all cases). There was a moderate correlation between $TT_1$ and $VT_1$ ($\rho = 0.41$, $p = 0.021$) and a strong correlation between $TT_2$ and $VT_2$ ($r = 0.78$, $p < 0.0001$). In Fig. 5, rectal temperature data for all participants are presented in relation to the relative running speed, expressed as a percentage of maximal running speed (averaged at 2% steps), to visualize and confirm the presence of three CBT domains during incremental running and the similarity of breakpoints between them with the gas exchange thresholds.

The regression analysis for comparison of temperature with gas exchange thresholds is shown in Fig. 6.

The agreement between $TT_2$ and $VT_2$, tested by the Bland-Altman method, was satisfactory. After exclusion of an outlier (participant no. 8), the mean difference between $VT_2$ and $TT_2$ was $-0.47 ± 1.24$ km/h, with 95% LoA $-2.90–1.97$ km/h. In 68% of participants the differences between $TT_2$ and $VT_2$ were within ± 1 SD of the mean difference (*i.e.,* within $-1.71–0.77$ km/h range). Only in four participants (12.5%) the differences were greater than 1.5 km/h.

## Test-retest reliability

Test-retest reliability for threshold values expressed as running speed was low (ICCs for $VT_1$, $TT_1$, $VT_2$, and $TT_2$ were 0.39, 0.18, 0.49, and 0.23, respectively). In some cases, the evaluators indicated a first-choice and an alternative (third) breakpoint/threshold value. In all such cases the indicated values matched among evaluators but were not always designated as the same level of choice. When these values were matched and included in analysis, the test-retest reliability was higher for $VT_2$ ($ICC = 0.93$) but still showing poor reproducibility for $TT_2$ ($ICC = 0.36$). The test-retest reliability was generally higher for

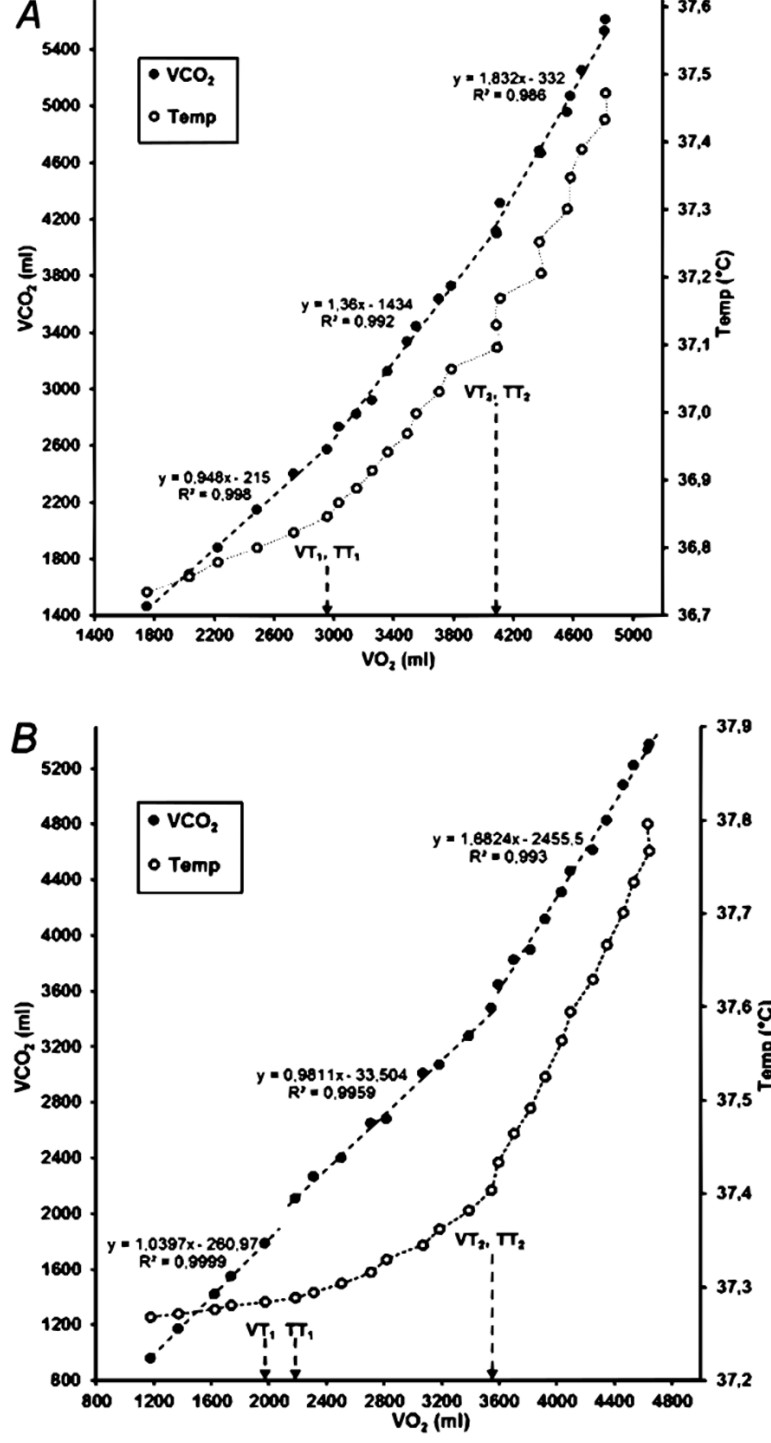

**Figure 4  VO$_2$ in relation to VCO$_2$ (V-slope) and rectal temperature in two participants.** Note that the marked temperature thresholds were determined from the temperature data in relation to running speed (see Fig. 1). (A) Identical gas exchange and temperature thresholds; (B) TT$_1$ is one stage (0.5 km/h) higher than VT$_1$.

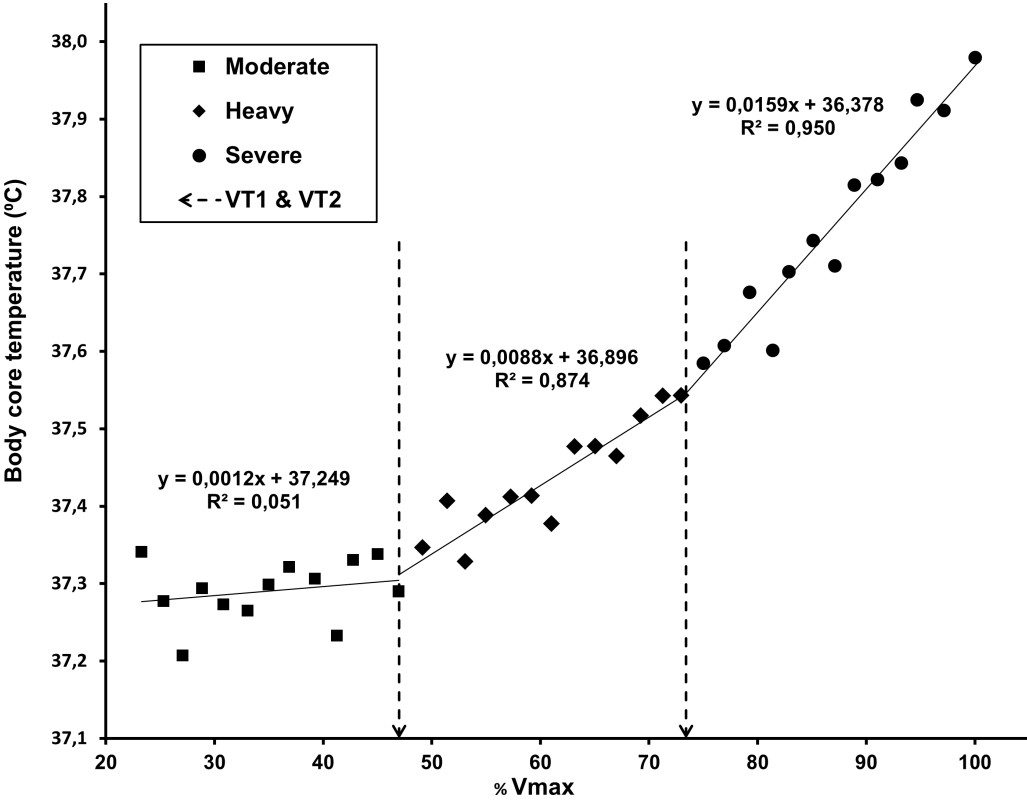

**Figure 5** **Core body temperature data for all participants, averaged in relation to the percentage of maximal running speed (2% steps).** The linear regressions for the three domains were calculated with average $VT_1$ and $VT_2$ values as breakpoints (dashed lines).

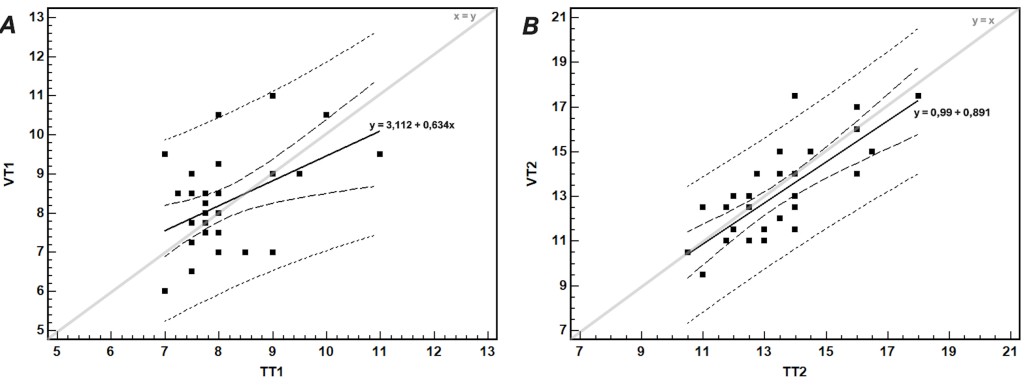

**Figure 6** **The relationship between (A) the first gas exchange ($VT_1$) and temperature ($TT_1$) thresholds, and (B) the second gas exchange ($VT_2$) and temperature ($TT_2$) thresholds.** The thick continuous line shows the linear regression with corresponding equation; the short- and long-dashed lines show the 95% prediction and confidence intervals, respectively. The thin line shows the identity line.

threshold values expressed as $VO_2$ (ICC = 0.52, 0.72, 0.69 and 0.68 for $VT_1$, $TT_1$, $VT_2$ and $TT_2$, respectively).

## DISCUSSION

The aim of this study was to model the changes of core body (rectal) temperature during graded treadmill running to volitional exhaustion and to evaluate the relationship between CBT pattern with concurrent ventilatory and gas exchange pattern. The results confirmed our hypothesis that rectal temperature in young, fit men increases disproportionately with increasing running speed, showing threshold-like changes. Moreover, in most cases, two breakpoints in the CBT-speed relationship are present, and those thresholds are significantly correlated ($TT_1$, moderately; $TT_2$, strongly) to the first and second gas exchange threshold ($VT_1$ and $VT_2$, respectively).

The increase of rectal temperature in most of our participants followed a 3-phase model, while a two-segment model was detected in only three of our participants (Fig. 3), showing a single breakpoint that occurred at ~70% of the maximal attained intensity (Figs. 3B, and 3C), similar to esophageal temperature thresholds for $VE/VO_2$ and $VE/VCO_2$ in the studies of *White & Cabanac (1995)* and *White & Cabanac (1996)*.

Baseline temperature (37.20 ± 0.27 °C) and the highest temperature measured during the recovery of our participants (38.42 ± 0.38 °C) were also comparable, although higher, than the range of esophageal temperatures registered in the previous studies (*White & Cabanac, 1995*; *White & Cabanac, 1996*; *Sancheti & White, 2006*). Lower readings of esophageal *vs* rectal temperature during exercise in different environmental conditions were also described by previous studies (*Mündel et al., 2016*). A rectal temperature dip was observed in some of our participants at the beginning of the exercise. Small esophageal (blood) temperature dips (−0.2 °C) at the very beginning of a cycle ergometer test (*White & Cabanac, 1995*; *Sancheti & White, 2006*) were previously described and interpreted as a result of: (1) an increased limb perfusion causing increased return of cold blood from cutaneous circulation to the core, and (2) the heat produced and retained in active muscles for intramuscular warming, rather than being transported away from the muscles by blood (*Alt et al., 1986*; *Alt, Stangl & Theres, 1993*). Arguably, the stagnant or slow CBT rise in the moderate intensity range observed in some participants can be partly attributable to the lower temperature in the working muscles at the start of a graded running test, and the time needed for heat accumulation in the active muscles and equalization of muscle and core temperatures.

In the study comparing CBT with ventilatory parameters during graded cycling to exhaustion *White & Cabanac (1995)* described a disproportionate increase of both $VE/VO_2$ and $VE/VCO_2$ in relation to CBT. This onset of hyperventilation proportional to the increase in CBT at higher exercise intensities was described as a thermolytic response for selective brain cooling, from enhanced upper-airway evaporation and heat loss (*White & Cabanac, 1995*; *White & Cabanac, 1996*; *Sancheti & White, 2006*). Unfortunately, the authors did not report the CBT/workload relationship, hence giving no real evidence on the occurrence of a core temperature threshold with increasing exercise intensity. Moreover,

the reported CBT threshold values were dependent upon the testing protocol (fast *vs* slow workload increase) and temperature probe placement (tympanic *vs* esophageal).

The premise that hyperventilation during high intensity activity contributes to selective brain cooling has also been contested, as decreasing arterial $CO_2$ pressure (at intensities above AnT) contributes to cerebral hypoperfusion, thus diminishing heat removal and likely increasing brain temperature (*Kety & Schmidt, 1948*; *Nybo et al., 2002*; *Nybo & Secher, 2011a*; *Nybo & Secher, 2011b*). Therefore, it seems that metabolic factors, rather than brain cooling, primarily drive the disproportionate increase of both core temperature and ventilatory indexes during incremental exercise, following increased recruitment of fast twitch motor units at higher intensities (*Nagata et al., 1981*; *Moritani et al., 1984*; *Airaksinen et al., 1992*; *Mateika & Duffin, 1994*; *Chwalbińska-Moneta, Hanninen & Penttila, 1994*; *Taylor & Bronks, 1994*; *Lucía et al., 1997*; *Lucía et al., 1999*; *Hug et al., 2003*). *Aaron et al. (1992)* measured the $O_2$ cost of hyperpnea during progressive exercise in healthy young subjects. From moderate to severe exercise, they noted an $\sim$80% increase in the oxygen cost per L of ventilation, while the average increase in total $VO_2$ per step devoted to ventilation increased fivefold, from 8% to $39 \pm 10\%$. The results of this study indicate that the decrease of ventilatory efficiency with increasing workload may considerably contribute to steeper CBT rise and the appearance of CBT thresholds. *Meyer et al. (2004)* demonstrated that metabolic acidosis induced by exercise is causally involved in the occurrence of hyperventilation at the respiratory compensation point ($VT_2$). However, the $VT_2$ in their study occurred even with complete buffering of metabolic acidosis, indicating that other physiological stimuli are included in the regulation of hyperpnea during intense exercise to volitional exhaustion.

Expressed as % $VO_{2max}$, the average values of $TT_1$ and $TT_2$ in our participants were $56.8 \pm 7.1\%$ and $83.3 \pm 10.0\%$ $VO_{2max}$, respectively. The average observed value was somewhat higher for $TT_2$ than $VT_2$ (Table 1), congruent with the results of *Lucía et al. (1999)* who reported higher values of the second EMG threshold for both *m. vastus lateralis* ($86.9 \pm 1.5\%$ $VO_{2max}$) and *m. rectus femoris* ($88.0 \pm 1.4\%$ $VO_{2max}$) compared with the values of $VT_2$ ($84.6 \pm 6.5\%$ $VO_{2max}$). No statistically significant differences between the thresholds were found in that study ($p > 0.05$) (*Lucía et al., 1999*).

The objectivity of evaluators in determining $TT_2$ was high, and threshold values were clearly identifiable in most participants. However, the existence of $TT_1$ could not be firmly sustained due to poor agreement between the evaluators, as well as due to several cases (22%) in which this threshold could not be identified. In the study by *Lucía et al. (1999)* the first EMG threshold was identifiable in all 28 participants, although the authors did not state the level of agreement between evaluators in identification of the threshold. The low objectivity of the evaluation of $TT_1$ indicates that the aerobic threshold cannot be reliably assessed from changes in CBT during a treadmill GXT. This might be influenced by the very low variability of $VT_1/TT_1$ and by discrete metabolic changes taking place at lower workloads, producing an unfavorable signal-to-noise ratio. In addition, the difficulty of $TT_1$ assessment could result from the fact that the intensity at the aerobic threshold corresponds to the transition speed between walking and running gaits (*Sentija & Markovic, 2009*), a speed naturally avoided in locomotion (*Minetti, Ardigo & Saibene, 1994*).

The good agreement between $TT_2$ and $VT_2$ is comparable to the LoA ($\approx$ 2–2.5 km/h) found between the second EMG and ventilatory thresholds (*Lucía et al., 1999*) as well as between the lactate and ventilatory thresholds ($\approx$ 2–3 km/h) in the study of *Gaskill et al. (2001)* (approximate values, recalculated from original values reported in watts (*Lucía et al., 1999*) and ml $O_2$/min (*Gaskill et al., 2001*)).

The mean difference between $TT_2$ and $VT_2$ (0.48 km/h, $p = 0.029$, Table 1) converted to time difference between the two equals 28.2 ± 74.4 s, with $TT_2$ lagging behind $VT_2$. This statistically significant difference between thresholds, defined as running speed, is negligible from a practical standpoint. $TT_2$ and $VT_2$ are two different indicators supposedly originating from the same underlying physiological processes and the time lag in their onset might be the consequence of their sequential occurrence and/or data latency. With increasing running speed, changes in some ventilatory variables like minute ventilation, tidal volume and breathing frequency are registered almost instantaneously. On the other hand, the change in CBT kinetics (increased rate of heat production within the active muscles) is registered with a delay for the blood transit time from the muscle capillaries of the locomotor and respiratory muscles to the temperature collection point (*Dempsey, Harms & Ainsworth, 1996*; *Kalliokoski, Knuuti & Nuutila, 2004*).

The high correlation between $TT_2$ and $VT_2$ is congruent with the results of previous studies that compared different methods for threshold detection, such as the study by *Tikkanen et al. (2012)*, showing a high correlation between the EMG threshold and $VT_2$ ($r = 0.86$, $p < 0.001$) and between the EMG threshold and the onset of blood lactate accumulation ($r = 0.84$, $p < 0.001$). *Kang et al. (2014)* compared anaerobic thresholds estimated by EMG and ventilatory parameters using different data filtering intervals (9, 15, 20, 25, 30 s) and detected a high correlation ($r = 0.89-0.99$) for different combinations of EMG and gas exchange data filtering.

The test-retest reliability found in the present study is lower than previously observed for EMG thresholds (ICC range 0.73–0.96 for the first and second EMG threshold) (*Lucía et al., 1999*) and for thresholds determined by the Dmax method ($r = 0.78-0.93$) (*Cheng et al., 1992*). On the other hand, our results are comparable to those of *Dickhuth et al. (1999)*, who found a low reliability of epinephrine and norepinephrine thresholds ($r = 0.49$ and $r = 0.46$, respectively). The possible reasons behind the low test-retest reliability of the temperature thresholds determined in the present study include: small sample size; biological variability of the measured variables; error(s) caused by variation inherent to the measurement methods; error in subjective estimation (inter- and intra-evaluator), previously identified as potentially appreciable source of error (*Gladden et al., 1985*). Moreover, the results might have been influenced by the homogeneity of the sample of participants who repeated the test (showing above average fitness, and a narrow overall $TT_2$ range), and by the suboptimal running speed resolution in the test (0.5 km/h).

## Limitations of the study

The findings of this study can only be generalized in reference to the characteristics of the sample, and therefore the conclusions drawn from the data are limited to healthy, active young men. Nevertheless, since the proposed causal physiological mechanism

behind the occurrence of the thresholds (a steeper temperature increase at the transition from moderate to heavy, and from heavy to severe intensity) is metabolically driven and therefore supposed to be ubiquitous, we may presume the appearance of the temperature thresholds regardless of the participants' characteristics. The study also failed to account for thermoregulatory factors like sweating and skin temperature regulation. To improve research findings, further studies should ensure representative samples with a broader range of participants, including both genders, various age groups and different fitness levels. The intra-evaluator reliability in detecting the temperature thresholds should also be tested.

Another methodological limitation of this study refers to the anatomical location for collection of CBT data. The use of tympanic and/or esophageal temperature was recommended in previous studies, as measurement sites more closely reflecting core (brain) temperature (*White & Cabanac, 1995*; *White & Cabanac, 1996*; *Sancheti & White, 2006*; *Lim, Byrne & Lee, 2008*), and, arguably, esophageal temperature also better reflected the increased passage of warm air through the adjacent airways. In line with our hypothesis, rectal temperature should more closely reflect the intensity-related increase in CBT, with a fast response due to the proximity of large pelvic and thigh muscles active in running, both *via* conductive (solid tissue) and convective (blood) heat transfer. The depth of insertion of the temperature probe was shallower than recommended by *Hymczak et al. (2021)*, as it was chosen not to represent the best indication (highest value) of internal body temperature, but rather to be most reflective to rapid core temperature changes during incremental running. As shown in the study by *Lee et al. (2010)*, $T_{re}$ at shallower depths is most reflective of those rapid core temperature changes. Even so, and regardless of the strict procedure for insertion and securing of the rectal temperature probe, and the instructions given to the subjects regarding the depth of insertion ($\sim 8$ cm beyond the anal sphincter), we cannot exclude possible minor displacements of the temperature probe within the rectal area during measurement. Therefore, the location and variation of the depth of the rectal probe insertion may have contributed to a certain amount of variability of the measured temperature values (*Lee et al., 2010*) and the derived parameters.

## CONCLUSIONS

In conclusion, graded treadmill running induces a disproportionate increase in rectally measured core body temperature, with detectable breakpoints moderately ($TT_1$) and highly ($TT_2$) related to the first (aerobic) and second (anaerobic) ventilatory thresholds. Different patterns of rectal temperature increase were registered, with a 3-phase segmented linear regression model as the most common. Overall, smoother between-phase transitions and a higher signal-to-noise ratio was noted for CBT data, than for breath-by-breath gas exchange parameters. Highly objective assessment of the $TT_2$, satisfactory agreement and correlation between $TT_2$ and $VT_2$ imply the presence of a temperature threshold that may be used in estimation of the anaerobic threshold. On the other hand, $TT_1$ showed unsatisfactory reliability, lower objectivity and correlation with $VT_1$. We presume that the same underlying physiological mechanisms account for the occurrence of gas exchange and

core temperature thresholds with increasing workload. Namely, a decrease in locomotor and ventilatory efficiency, due to sequential motor unit recruitment pattern and hyperpnea, are manifested as thresholds delineating the moderate, heavy and severe intensity domains of physical activity. Further studies should be considered to elucidate the low reproducibility of the temperature thresholds described in this study, and to improve the methodology and enable practical implementation of core body temperature measurement for demarcation of exercise intensity domains. Future studies should also investigate the comparison of the temperature thresholds with other parameters used to assess the anaerobic threshold (EMG, blood lactate, *etc.*), and whether the findings in this study can be translated to diverse populations in regard to sex, age, fitness and acclimatization level, to modalities of graded exercise other than running (*i.e.,* cycling, walking, rowing, *etc.*), and test protocols of different total duration and workload increment.

**Abbreviations**

| | |
|---|---|
| **AeT** | aerobic (or lactate) threshold |
| **AnT** | anaerobic threshold |
| **BF** | breathing frequency |
| **CBT** | core body temperature |
| **CO$_2$** | carbon dioxide |
| **EMG** | electromyography |
| **GXT** | graded exercise test |
| **ICC** | intraclass correlation coefficient |
| **RCP** | respiratory compensation point |
| **T$_{re}$** | rectal (core body) temperature |
| **TT$_1$** | first temperature threshold |
| **TT$_2$** | second temperature threshold |
| **TV** | tidal volume |
| **VE** | minute ventilation |
| **VCO$_2$** | carbon dioxide output |
| **VO$_2$** | oxygen uptake |
| **VT$_1$** | first gas exchange (ventilatory) threshold |
| **VT$_2$** | second gas exchange (ventilatory) threshold |

## ACKNOWLEDGEMENTS

The authors would like to acknowledge and thank all the study participants, and Prof. Vlatko Vučetić, University of Zagreb Faculty of Kinesiology, for his assistance during the data collection.

### Funding

The study was supported by grant from the Croatian Ministry of Science, Education and Sport, Grant Number: 034-0342607-2279. Recipient: Davor Šentija. There was no additional external funding received for this study. The funders had no role in study design, data collection and analysis, decision to publish, or preparation of the manuscript.

## Grant Disclosures

The following grant information was disclosed by the authors:
The Croatian Ministry of Science, Education and Sport: 034-0342607-2279.

## Competing Interests

The authors declare there are no competing interests.

## Author Contributions

- Marija Rakovac conceived and designed the experiments, performed the experiments, analyzed the data, prepared figures and/or tables, authored or reviewed drafts of the article, and approved the final draft.
- Davor Šentija conceived and designed the experiments, performed the experiments, analyzed the data, prepared figures and/or tables, authored or reviewed drafts of the article, and approved the final draft.
- Tošo Maršić conceived and designed the experiments, performed the experiments, analyzed the data, authored or reviewed drafts of the article, and approved the final draft.
- Vesna Babić performed the experiments, authored or reviewed drafts of the article, and approved the final draft.

## Human Ethics

The following information was supplied relating to ethical approvals (i.e., approving body and any reference numbers):

University of Zagreb School of Medicine and Faculty of Kinesiology Review Board (approved research proposal 04-3741/2-2008). The research was approved within the dissertation proposal defense procedure.

## Data Availability

The data are available in the Supplementary File 'Data - evaluated thresholds'. The file shows the ventilatory and temperature threshold values identified by three independent evaluators. The values were used in the evaluation of the relationship between core body temperature pattern and concurrent ventilatory and gas exchange pattern during a graded treadmill test, as described in the manuscript.

## Supplemental Information

Supplemental information for this article can be found online at http://dx.doi.org/10.7717/peerj.19686#supplemental-information.

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
