# Peer review of "Core body temperature correlates of transition from aerobic to anaerobic metabolism in running"

_PeerJ, doi:10.7717/peerj.19686_

## Round 0.1 · original submission · Major Revisions

Please make appropriate changes according to reviewers' comments or write a detailed rebuttal on a point-by-point basis.

Reviewer 1 ·

Basic reporting

This study had 30 male young and trained participants performing a graded exercise stress test, while measuring gas exchange and core body temperature. The aim of the study was to identify the aerobic and anaerobic thresholds utilizing the development of core body temperature and compare this to the more traditional gas exchange thresholds. This is proposed to be a rather straightforward study design, and performed as such, but some major limitations remain and need discussion. Therefore, I suggest major revision, which are summarized below, followed by minor comments. The minor comments are mostly regarding language and content. The scientific language could be improved, and the content is sometimes redundant while other times, important explanations are missing. I overall like the study, but I believe that there needs to be a more critical look on using temperature thresholds and a more detailed exploration of thermoregulatory factors besides metabolic heat production. At the moment I believe that the use of the TTs sounds a bit too promising. The figures need better resolution. (See detailed major and minor comments in "4. Additional comments").

Experimental design

The experimental design is good and meets the expactations.

Validity of the findings

The rational (i.e., theoretical background) or justification of the research hypotheses needs to be improved. The Conclusions need to incorporate more discussion and criticism of the temperature thresholds (see major and minor comments in 4. Additional comments).

Additional comments

Major comments:
The overall tenor of the manuscript is that the authors argue that the temperature thresholds are more or less correlating with the gas exchange thresholds, although the authors mention major limitations themselves. The biggest limitation of using temperature instead of gas exchange in practice (and as shown here in this study), is the low reliability of forming the temperature thresholds. In my opinion, this low reliability is linked to two factors (1 and 2 below), that both need more discussion.

1) Explain more details how hard it is to identify the temperature threshold.
Looking at figure 1 showing the curve of core body temperature it is not surprising that it is hard to find thresholds in this curve. I would like to see further explanation on why the 3 evaluators did sometimes not agree and whether the same evaluators would judge the same curve differently on separate occasions. This is discussed to some extend but not enough. Is it really a practically usable method?

2) Discuss other factors that influence heat strain and how/if they are hypothesized to change the thresholds?
The variability of and different influences on core body temperature need to be discussed more as they may affect these thresholds. I agree that core body temperature and heat strain, are strongly linked to the metabolic rate, but also there are other factors that mediate an individual’s ability to cope with heat stress. The acclimatization status for example is not discussed at all. The fitness level of participants would play a role. Furthermore, the gender, the age, the health status, the body composition and fat mass, could for example influence heat strain. In the presented data, the correlation is only found in a very homogenous group (same sex, similar age, body composition and fitness level, etc.). This needs to be discussed as it is someone that would want to implement this method in practice would encounter different clients. Further it is not discussed how different climatic condition might influence these associations. Is there reason to believe these thresholds would hold true in different environments? I suggest some recent publications to check and implement some of their ideas:
- Cramer, MN, Jay, O. Biophysical aspects of human thermoregulation. Autonomic Neuroscience: Basic and Clinical. 196 (2016), 3-13.
- Bernard, TE, Wolf, ST, Kenney, WL, A Novel Conceptual Model for Human Heat Tolerance. Exercise and Sport Science Reviews, 2023, DOI: 10.1249/JES.0000000000000332

3) Fix up introduction: Improve justification/ theoretical background of study. Why did you think about testing this in the first place?
- Line 91: I think before you go into the theory you have to set the scene. You describe other methods to determine thresholds very briefly in the previous paragraph.. maybe then this one could start with: One other proposed method to determine thresholds is the measurement of CBT. This is because, it has been shown…
- I am not sure why there is an extra paragraph on the EMG method and all others are just mentioned without explanation. I don’t think this paragraph on the EMG method is needed, unless there is a clear link to the CBT method. If the latter is the case this needs to be made clearer as I don’t understand this link as presented here.
- Line 91 -93: this sentence is not enough to explain your theoretical background. I think you should reduce the EMG paragraph and extend this explanation of the theory that is important for your study. At the moment you have one sentence explaining the theory and then jump to your hypothesis, that’s not enough.
- Line 102: “may coincide with changes in ventilatory parameters” it really sounds like you are just guessing that this could be the case. What are the reasons you make this assumption?
- Also I think addressing major comment 2) will lead to a new paragraph in the introduction.

4) Fix up discussion: As mentioned above, the discussion needs to be more critical reflecting the findings and not overselling the threshold as possible replacement for VT.
This will be shaped by a) how you change the introduction and b) how you incorporate more thermoregulatory theory.
Finally, I think your conclusion must be that, at the moment, and based on this study’s findings, you cannot suggest using CBT to determine anaerobic and aerobic thresholds. You may hypothesize which future developments and studies might be needed to achieve this. But a test with this low reliability although performed with such a homogenous group and the presented issues that evaluators did not identify the same thresholds, needs to be presented as “not ready for practical use” in my opinion.





Minor comments:

Abstract:

- Line 17: I think purpose could be improved. Why did you do this comparison?
- Line 20: Ten participants repeated the test for reliability assessment. If you have that in the Methods of the Abstract, I will also suggest that you include the outcome of this, namely that it is not so reliable. At the moment, it sounds misleading: If I read that “we tested reliability” but no comment on the outcome I assume that it was good reliability. So, this needs to be explained that it was not.
- Line 23: please define the criteria here also or briefly sum it up. Referencing how you do it later is not so good.
- Line 31: TT1 and VT1 were moderately correlated (p = …); used p twice instead of r
- Line 32: It is not clear what the statistical difference is describing here. TT1 vs TT2?

Introduction:
- Line 52: change word inexorably
- Line 79: remove etc. and put or before self-reported measures.
- Line 86: increased instead of increases
- Line 81-89: why is this paragraph on the EMG method here? Doesn’t seem fitting or logical to me. Couldn’t it just be mentioned briefly together with the methods in the previous paragraph?
- ADD: I think there is a paragraph missing on explaining what other factors contribute to CBT. See major comment 2 and 3.

Methods:
- Line 134. How were these participants recruited? It seems that this is a rather homogenous group of participants.. Was this a specific cohort, i.e. students or sport students or just general public? Why were no females included? And what was the seasonal acclimatization of the participants?

- Air temperature at 22 and RH<60. Could it be that if you repeat this test at other temperatures these thresholds change? I am wondering if the practical application of this method would only be given if this holds true at different environmental conditions and with more heterogenous groups.. i.e. older younger, heavier, females, less fit, less acclimatized.

Results:
- Line 251: Why you go straight from describing the first moderate domain to the recovery period?
- Line 253: Not really a high CBT.. How do they explain that?
- Line 255 following. This belongs to the limitation section in the discussion not here.
- Line 255: also, not say “our participants” just “four parcticipants”.
- Line 285: there is a significant difference between running speed at VT2 and TT2, but a very high correlation between VT2 and TT2? I don’t always understand what you are comparing or correlating this needs to be made clearer.

Discussion:
- Line 327: didn’t you say in the introduction that there were no comparable studies that tested this?
- Line 381: The link to the EMG threshold was not introduced clearly in the introduction. So, I don’t understand this link.
- I am missing a discussion of other factors that can influence CBT! And the value of this in a practical setting where temperatures inside a lab may change and heat stress tolerability might change (already depicted by the poor re-test reliability).
- I am also missing a discussion regarding the overall quite low CBT values. Recent studies often found CBT of up to 40 degrees in participants. Could this be linked to the short time spend in each zone? Your test overall lasts only 12 minutes.. what if a different form of a graded test would be used (i.e. 3 min per stage, would an extended exercise duration increase CBT and how would this influence TTs)?

Figures:

- FIGURE 1: Wow, this looks like a pretty linear increase to me.. not sure how you identified thresholds there. Seeing this I am not surprised that not all three evaluators were finding the same thresholds, and that the method was not so reliable.
- All Figures need to be in better resolution.
- Table 1: So, from VT1 to VT2 there is less than 0.3 degree increase in Core Body Temperature.? And at the end CBT is only 38 degrees.. I think it should be further discussed why these are such low values, given that in recent years in exercise athletes have often been found to reach CBT of up to 40 degrees. Could this be linked to the short duration of the steps, i.e. exercising longer at the same intensity would lead to an increase in CBT even at lower speeds?

Other comments:
It seems like you presented this data at ECSS 2012 (https://www.croris.hr/crosbi/publikacija/prilog-skup/588270) and 2013 (https://www.croris.hr/crosbi/publikacija/prilog-skup/598448). Is there any particular reason why you decided to submit this data for publication more than 10 years after the initial presentation?

Reviewer 2 ·

Basic reporting

English is not ambiguous. Just some phrases need to be reworded.
1. Line 117…: paragraph contains repetitions. I think it could be reorganized to finish with the hypothesis. (Line 129: neither…nor?)
2. Line 140: is review board an official ethics committee.

Considering to the background, I think it is a good point. I would like to discuss some point and look after your opinion
3. Line91: Independently from lower thermodynamic efficiency, I think other mechanisms are involved. If type II fibre consume more ATP, their higher power production can make this difference smaller and size effect may be discussed. Can it be considered that ATP oxidative resynthesis is the most heat producer in comparison to other pathways (PCr and glycolytic)(González-Alonso, Quistorff, Krustrup, Bangsbo, & Saltin, 2000). Muscle and whole body temperature is the result of heat production and dissipation balance. In a context of exercise induced hyperthermia, how do you take into account the thermoregulation effectors notably with unsteady sweating onset, interfering with heat production? Do you consider it is almost only metabolic issue?
4. Figures quality
4.1. Figure 1: Could the quality of the figures be improved? The image resolution seems questionable. The curve 1-A seems really noisy. Why did you use different scales for A and B, C, D? For clarity of reading, may I propose to standardize the scales?
The threshold may be pertinent to be added on at least one figure with speed on x-axis. Why did you not report same graph with measured core temperature expressed as a function of the speed?
In the legend, I find that the mention of “participant n15” may induce confusion. Maybe It could be added something like “Core body temperature for one subject” or somewhat to precise that it is an illustration of your approach (like figure 2).
4.2. Figure 5: I do not understand the information provided. Is core temperature averaged by phase? Why do we have 13 points by phase? Can you explain how do you build this figure?
4.3. Would a graph of mean of all subject CBT as a function of relative speed be pertinent? This may provide information about global thermoregulation response to this incremental test.
4.4. To appreciate the level of interindividual variability would it be pertinent to draw TT in a Bland-Altman like mentioned in statistics or box plot with the difference between TT and VT for 2 thresholds or somewhat?

Experimental design

This work involves a really interesting and original topic. The study explores potential link between ventilatory thresholds and exercise induced hyperthermia, during graded exercise test. And finally, authors assume that CBT may be a new approach to determine the VT1 and VT2. I think the original research gap was identified.
1. My major comment is that even if the research gap is exposed, the question could be further clarified. I address the authors about the objective which is explored in this study. Is exploration of CBT response to graded exercise or the link between CBT kinetics and gas exchange? I would like to propose to define a main objective and secondary objective. I think it is important because, I feel like results and graphs are focused on CBT and gas exchange, even if we have one graph of all subjects CBT (fig 5).
2. Line 61-69: I think it is a global comment for the article but interesting to do here. I don’t understand if you include the anaerobic transition in the hypothesis of fibre type composition. May I propose that anaerobic threshold be more distinguished from anaerobic metabolism? It is an open discussion. That is a controversial point and I know concepts are moving. But as your hypothesis relies on gas exchange, would you agree with the fact that gas exchange measurement to determine anaerobic threshold reflects more acid-base status than metabolism switch to anaerobic pathway?(Connett, Gayeski, & Honig, 1986; Richardson, Noyszewski, Leigh, & Wagner, 1998) I think it is close to what you describe line 57-59. Thus, beyond the fact that the term of anaerobic threshold is inappropriate, do you think that more could be discussed about this controversy?
3. Line 138: Did you standardized breakfast composition in substrates composition and energy intake? Even if breakfast was ensured to be taken 2h before, dietary induced thermogenesis could have been taken into account(Westerterp, 2004).
4. Line 134: Were the participants different for morphology and sport activity?
5 Line 144: Did you standardize clothes for all participants?
6. Line 154: depending on subjects clothing, did the climatic conditions allow thermoneutrality period to assess resting core temperature? I think this is a major concern, given that core temperature was your principle measurement.
7. Line 159: Which Turbine size did you use?
8. Line 177: why do you not use lactate measurement.? As exposed in introduction, this may have improved the thresholds detection? And ventilator thresholds represent your reference value.
9. Line 187: I understand that probe was inserted at 8cm from anal marge to get better core temperature reactivity, but one of study justification is to provide a new criterion for threshold determination. Why do you not remain with classic rectal probe (10 to 15 cm from anal marge) or oesophageal? I mean that the accuracy was a critical point but also for the transposition of results, by taking into account common practice to monitor core temperature.
10. Line 191: Visually method? Not objective criterion like derivation or percentage of variation. How do you define accelerated increase of CBT? Variation percentage? Ultimately, the kink point for CBT was coupled to time? Or Absolute speed?
11. Line 196: Did you try other method by machine learning or Kmeans for example to define kink points?
12. Line 222: paired T-test were performed on which value for TT (Temperature, V ̇O2, speed) and VT (speed or V ̇O2)
13. Line 232.No participants were excluded? (line264 and line 300). Was Participant 8 used for regression?
14. Line:243: do these 3 participants have the two thresholds defined?
15. Line 245: do you observe subject’s characteristics which could explain this difference in regression model?
16. Line 248: This discrepancy in Tre at the beginning of the exercise. raises the issue of thermoneutrality measurement. Without a clearly defined thermoneutrality period, it remains difficult to interpret this initial response.
17. Line 268 What can you conclude about this observation
18. Line 281 differences between TT1 and VT1 are assessed by running speed?
19. Line 342: Maybe, phenomenon of vasomotor regulation may be involved in exercise beginning? Exercise was shown to induce vasoconstriction at the beginning and, furthermore, to increase core temperature threshold for vasodilation threshold (Kellogg, Johnson, & Kosiba, 1991a, 1991b) Furthermore, skin temperature is an important factor in vasomotor control. This measurement could have provided some information about core temperature regulation. (John M. Johnson & Dean L. Kellogg, 2010; J. M. Johnson & Kellogg, 2010)
20. Line 411 “same physiological process”: can you clarify what kind of physiological processes? If there is an underlying metabolic mechanism, which hypothesis do you propose?
21. Line 440-…: I think lactate measurement may have allowed better reproducibility, even though correlations are satisfying for TT2.
Limitations about thermophysiological assessment may be added. The correlation between TT and VT does not mean common physiological pathway. To ensure about the reproducibility of the result, I think we should be able to assess the thermoregulatory response which includes not only core temperature but also, at least, skin temperatures, sweating (threshold, volume) and energy expenditure.
22. Line 461(same comment than point 9) I understand the reason of methodological choice shallower Tre is reflecting heat from redistribution but objective is core temperature. I think Tre was used to monitor core temperature. Classic installation may be interesting because reliable core temperature is a major concern. All the more than the aim was to propose another objective criterion for threshold assessment.

Validity of the findings

This study provides interesting insights for improving the determination of intensity transition of exercise during a graded exercise test. A dominant 3-phase pattern was identified and may allow to a new tool for threshold determination. Notably, the strong correlation between TT2 and VT2, along with a better signal-to-noise ratio, supports the initial hypothesis of a relationship between gas exchange and core temperature.

23.The authors provide statistical analysis which allow to conclude about results. I think additional figure could be useful for the understanding of results, like Bland-Altman or somewhat.
24. For the objective which involves the description of core body temperature response to graded exercise test, additional data could be necessary if we wanted to explore.
25. Conclusions of the authors are well supported by the results. This study allows to make correlation but not to explore underlying mechanisms. However, different patterns in kinetics of core temperature and the lack of reliability for VT1 leave many questions unanswered. We would have appreciated some insights on these results.
26. Line 479-483: what can you propose in the study to support the underling mechanisms you proposed to explain these results? To introduce research perspectives, considering to the kinetics of the core temperature in the graded exercise test, thermoregulatory aspects should be addressed to take into account the onset kinetics of thermolysis effectors, the energy expenditure, and the reproducibility of the response. If there is a common mechanistic link between core temperature kinetics and ventilatory response, we can assume that ventilatory variable may be altered by situation like heat acclimation, which could be non-metabolic induced alteration of this variables(Beaudin, Clegg, Walsh, & White, 2009)

Additional comments

This is a big challenge because it is a condition where several elements have to be taken into account in a situation of unsteady regulation (physiological hyperthermia before stabilisation): muscle heat production (absolute intensity, type fibres composition and recruitment, exercise modality, metabolism contribution, substrate use profile), thermoregulation, morphological characteristics. I suggest that context be more defined, to present current issues about thermophysiological challenges and ventilatory threshold. Given the title of the article, if aerobic anaerobic transition is mentioned, clarification must be added to what extent ventilatory threshold 2 can be associated to increase in heat gain. I think studies like Gonzalez & al. may be discussed {González-Alonso, 2000 #2927}

Beaudin, A. E., Clegg, M. E., Walsh, M. L., & White, M. D. (2009). Adaptation of exercise ventilation during an actively-induced hyperthermia following passive heat acclimation. Am J Physiol Regul Integr Comp Physiol, 297(3), R605-614. doi:10.1152/ajpregu.90672.2008
Connett, R. J., Gayeski, T. E., & Honig, C. R. (1986). Lactate efflux is unrelated to intracellular PO2 in a working red muscle in situ. J Appl Physiol (1985), 61(2), 402-408. doi:10.1152/jappl.1986.61.2.402
González-Alonso, J., Quistorff, B., Krustrup, P., Bangsbo, J., & Saltin, B. (2000). Heat production in human skeletal muscle at the onset of intense dynamic exercise. The Journal of physiology, 524 Pt 2(Pt 2), 603-615. doi:10.1111/j.1469-7793.2000.00603.x
Johnson, J. M., & Dean L. Kellogg, J. (2010). Thermoregulatory and thermal control in the human cutaneous circulation. FBS, 2(3), 825-853. doi:10.2741/s105
Johnson, J. M., & Kellogg, D. L., Jr. (2010). Local thermal control of the human cutaneous circulation. J Appl Physiol (1985), 109(4), 1229-1238. doi:10.1152/japplphysiol.00407.2010
Kellogg, D. L., Jr., Johnson, J. M., & Kosiba, W. A. (1991a). Competition between cutaneous active vasoconstriction and active vasodilation during exercise in humans. Am J Physiol, 261(4 Pt 2), H1184-1189. doi:10.1152/ajpheart.1991.261.4.H1184
Kellogg, D. L., Jr., Johnson, J. M., & Kosiba, W. A. (1991b). Control of internal temperature threshold for active cutaneous vasodilation by dynamic exercise. J Appl Physiol (1985), 71(6), 2476-2482. doi:10.1152/jappl.1991.71.6.2476
Richardson, R. S., Noyszewski, E. A., Leigh, J. S., & Wagner, P. D. (1998). Lactate efflux from exercising human skeletal muscle: role of intracellular PO2. J Appl Physiol (1985), 85(2), 627-634. doi:10.1152/jappl.1998.85.2.627
Westerterp, K. R. (2004). Diet induced thermogenesis. Nutrition & Metabolism, 1(1), 5. doi:10.1186/1743-7075-1-5

Annotated reviews are not available for download in order to protect the identity of reviewers who chose to remain anonymous.

---

## Round 0.2 · accepted · Accept

Dear Authors, your manuscript is now acceptable for publication in its current form.

Reviewer 1 ·

Basic reporting

The basic reporting has improved with the previous review, I have no further remarks, well done.

Experimental design

The authors have answered all my requests, I have no further remarks.

Validity of the findings

I explained my previous concerns to the authors and the editor. The authors responded to all questions sufficiently, and the editor didn't seem to have similar concerns. Thereby, I have no further comments.

Additional comments

Thank you for adjusting your manuscript according to my suggestions and answering questions regarding my concerns. I think the quality of the manuscript has much improved, and I want to congratulate you on the study. From my side there are no further comments, I suggest to accept the article in its current form.